



# Peer-review of data products: an automated assistance system for INTERMAGNET

Roman Leonhardt[1], Benoit Heumez[2], Tero Raita[3], and Jan Reda[4]

[1]Conrad Observatory, GeoSphere Austria, Vienna, Austria
[2]Université Paris Cité, IPGP, CNRS, France
[3]Sodankylä Geophysical Observatory, University of Oulu, Oulu, Finland
[4]Institute of Geophysics, Polish Academy of Sciences, Warsaw, Poland

**Correspondence:** Roman Leonhardt (roman.leonhardt@geosphere.at)

**Abstract.** INTERMAGNET, a global network of geomagnetic observatories, publishes so-called "definitive" data products, which are subjected to an international peer-review system. Currently, geomagnetic data is submitted by about 100 observatories worldwide. Besides the mandatory one-minute (1-min) data products, INTERMAGNET has also accepted one-second (1Hz, 1-sec) data products for the past decade. The amount of data to be reviewed has significantly increased, making tra-

ditional manual data reviews increasingly challenging. The INTERMAGNET ROBOT (short IMBOT) has been developed to perform automated routines to convert and evaluate INTERMAGNET (IM) data submissions. The primary objectives of IMBOT are to (1) simplify one-second and one-minute data submissions for providers, (2) speed up the evaluation process significantly, (3) consider current IM archive formats and meta information , (4) simplify and speed up the peer-review process and finally, (5) reduce the workload of human data checkers. IMBOT automatically generates detailed reports and notifies

submitting institutes and human referees. It provides templates for corrections and also triggers re-evaluations automatically when data or any information in the submission directory is updated. This automated system makes data review faster and more reliable, providing high-quality data for the geomagnetic community.

## 1 Introduction

A peer review system is widely considered as essential to ensure the quality and accuracy of scientific research by allowing

experts in the field to evaluate and provide feedback before publication. Such reviews help to identify and correct errors, inconsistencies, or gaps in methodology, analysis, or interpretation, thus improving the overall reliability of the research. Although a peer-review system is widely used for scientific publications, pure data products are typically not reviewed by the science community. For about 20 years, INTERMAGNET has been using a system of mutual data checks on one-minute definitive data sets. As shown by MacMillan and Olson (2013), several geomagnetic data sets available at the World Data Center contained

errors such as unexplained discontinuities and drifts. This particularly affected submission before INTERMAGNET started its reviewing process in 2005. INTERMAGNET distinguishes "definitive" data (DD), subjected to an intense checking procedure, published on a yearly basis, and "real time" data, also called adjusted or variation products, useful for early warning systems and space weather applications. A group of volunteering data checkers is evaluating each "definitive" data submission. Initially



INTERMAGNET applied this peer review system to mandatory 1-min data products, which are used to evaluate the quality of observatory data and whether this observatory meets the standards of INTERMAGNET. Since 2014, INTERMAGNET welcomes submissions of data products with one-second resolution ($D_{sec}$). For effective archiving of such data sets, a new data format, IMAGCDF, has been introduced (Bracke, 2025). All INTERMAGNET observatories (IMO) are invited to submit these data sets along with their traditional one-minute data products ($D_{min}$). INTERMAGNET subjects submitted $D_{min}$ to a peer review system to ensure the quality and accuracy of published data through a two-step checking process. In the first step (step 1), an independent referee checks the data submission to identify errors, inconsistencies, missing elements, missing meta information, and evaluates the data against INTERMAGNET standards and thresholds. The console program "check1min.exe", executed automatically or manually, helps referees (Reda, 2021). In the second step (step 2), a member of the INTERMAGNET Operations Committee's definitive data group, usually the chair, cross checks the reports and data, comparable to the editor's decision in a peer-reviewed publication. If step 2 is successfully passed, the data are published on INTERMAGNET's web pages. The large amount of new $D_{sec}$ complicates this traditional approach. The acceptance of an observatory into INTERMAGNET remains solely based on the quality of its definitive $D_{min}$. Nevertheless, submitted $D_{sec}$ should also meet the high INTERMAGNET standards and their quality must be tested and evaluated through a transparent and conclusive process. Ideally, end users should have full access and understanding of the quality assessment procedure. A major challenge in evaluating $D_{sec}$ is the large amount of data, big file sizes, and limited software availability which complicate handling for data checkers. On the data provider side, similar challenges arise, including the need to create a new, sophisticated data format with previously unused meta information. These aspects are the main reasons why, for almost 10 years, all submitted $D_{sec}$ have not been reviewed.

In this article, we briefly summarize general aspects of a data reviewing process. We will list a number of reviewing tasks to be performed and discuss possible issues with data submissions, specifically submission of $D_{sec}$ to INTERMAGNET, although such issues might affect any other type of data submission. We introduce an automated routine written in Python to assist the peer-review process by taking over a significant amount of checking tasks. The principle idea behind IMBOT, the automatic data checker, is to minimize the workload for both data providers and data checkers, and provide high-quality data to end-users as early as possible. For demonstration and testing purposes, we analyse one-second definitive data submissions for two years, one shortly after introducing $D_{sec}$ to INTERMAGNET (2016) and a recent submission year for which the "call-of-data" deadline has passed (2022).

## 2 Data checking tasks

When it comes to scientific data products, the review process differs from that of scientific research, although there are some similarities. Generally speaking, a data product is a domain-specific, consumable entity aimed at transforming data into actionable insights for users. In order to accomplish this task, INTERMAGNET provides strict rules on data files, formats, contents, thresholds and meta information to be followed. Thus, the review process can be structured in a number of important bullet points to comply with these criteria. The role of data reviewing is to ensure that submitted definitive data meet INTERMAG-



NET standards in all of the required aspects and eventually, to provide recommendations for improvement. Minor corrections to meta information and file structure are possible from the referees side (as decided by the IM definitive committee in 2024), provided that approval is given by the data provider. However, modifications on data contents must be performed solely by the submitting entity. This procedure is similar to the review process of scientific publications, where layout changes and typographical errors can be corrected during the editorial process, requiring the acceptance (proofreading) of the authors. Any changes to scientific contents are made by the authors themself. Thus both automated and human reviewing processes need to preserve the original data contents.

## 2.1  Task 1: Verification of the correct amount of files and formats validity

The first reviewing task is the verification of the general contents of the submitted data product. This involves checking whether the data products contain the correct amount of data files, whether the naming principles are followed and whether the supplied formats are correct. For an INTERMAGNET $D_{min}$, the obligatory files comprise: 12 binary data files with monthly coverage, one ASCII data file containing baseline data, one README file, and one file containing yearly means for the observatory. Country information can be submitted but is not required for the initial submission. A $D_{sec}$ submission should include either 12 monthly files or 365/366 daily files in IMAGCDF format (Bracke, 2025). Currently, the submission of 365/366 daily files in IAGA-2002 format is also accepted. All requested files must be present in readable formats and follow the version specific naming conventions.

## 2.2  Task 2: Complete and appropriate meta information

Accurate meta information is the basis of all modern data acquisition. Metadata typically contains basic information on the acquisition location, site or station description, the sensor systems and their characteristics and data related parameters, like sampling rate and filter types. Metadata standards have been developed by many international networks in order to obtain coherent data products. Thus, a primary task of the reviewing process is to validate the provided meta information, which is part of the individual data files and associated README's. It is also necessary to check the consistency of the metadata across all files. The required meta information and its format description for INTERMAGNET is described in the technical manual (Bracke, 2025).

## 2.3  Task 3: Verification of data contents

It is necessary to check that each data file covers the projected time range and that missing data is marked with appropriate flags. It is also necessary to verify whether all components are present and that the corresponding columns contain appropriate data. Components as defined in the meta information, i.e. XYZ, DHZ in geomagnetic data, need to be present in the data files. Value ranges, resolution and units for each component need to follow the underlying format description. For geomagnetic data, vectorial information can be provided in various coordinate systems- spherical, cylindrical, cartesian- and be in different units. The correctness of this information and its consistency between files need to be checked.



## 2.4 Task 4: Verification of data consistency

Some data products contain averages and means derived from the same underlying data set. These averages need to be consistent within and across different files. The INTERMAGNET $D_{min}$ contains hourly and daily means within the binary data files, while yearly means are contained in the baseline and yearly mean files. Definitive $D_{sec}$ from a specific observatory need to be consistent with its corresponding definitive $D_{min}$, as both datasets sample the same local geomagnetic field at different frequencies. Thus, a filtered $D_{sec}$ needs to closely resemble the $D_{min}$. Besides inherent data consistency, it is also necessary to check the consistency of data products with the documented methodology. For example, geomagnetic activity, K indices, can be calculated in various ways (Menvielle et al., 1995); if a method is referenced, then the results need to be consistent with that methodology. Finally, data should also be consistent with a physical framework, which means that the variation records need to represent a real, unbiased record of the geomagnetic field. A common way to verify this is a comparison with well established data from a nearby location.

## 2.5 Task 5: Verification of data quality

The final task of the data review process concerns the evaluation of data quality. Overall, a unique measure of data quality is not easy to obtain in geomagnetic data as many typically used parameters -such as noise level, signal amplitudes- are strongly dependent on latitude, local geology, and proximity to oceans. Nevertheless, the data should be free of anthropogenic disturbances. Nearby magnetic disturbances are typically investigated by analyzing the difference between two sensors, often provided as delta-F between continuous vector and scalar measurements. Frequency disturbances can be assessed by power spectral analysis. Baseline values provide a measure of the long-term stability, and INTERMAGNET sets thresholds for acceptable long-term variations to below 5nT. Data continuity in the baseline requires steps to be below 1 nT between successive points. Changes in instrumentation or site characteristics might lead to larger "jumps" in baselines which should then be traceable and described accurately in the metadata. Data quality also comprises the timing accuracy. The local geomagnetic activity indices should resemble the global activity in a reasonable way, although no thresholds are defined for this comparison.

## 3 Basic concept of data reviewing supported by an automatic assistance system

When looking at the typical data reviewing tasks it is obvious that a significant proportion can be handled automatically. Such an automatic system needs to access data submission and then run a number of testing modules related to the tasks defined in the previous section. Ideally, an automatic process can also handle notifications and even referee assignments. The reviewing process of INTERMAGNET is separated in three steps described below, which are related to equally named data archives.

## 3.1 Step 1 - submission and review

Step 1 is related to the submission process by the data provider and includes any updates and/or corrections of submitted data products. Whenever data products are submitted to INTERMAGNET, they are uploaded to a geomagnetic information node



(GIN). Step 1 GIN is hosted by IPGP in Paris. Observatories/Institutes will obtain connection details after initial approval by INTERMAGNET officers. There are two different step 1 archives on the GIN, one for $D_{min}$, the other for $D_{sec}$. They have
a common structure, namely a yearly directory organization and will always host solely original raw data as uploaded by the data provider. IMBOT scans the directories daily, typically during the night (central european time), analysing whenever new data is uploaded or data has been modified. It will exclude data modified within the last two hours to prevent the analysis of unfinished uploads. After performing automatic tests, the ongoing procedure is slightly different between $D_{min}$ and $D_{sec}$.

– For $D_{min}$, the data provider and the assigned data checker will receive an automatic notification including a review
report generated from IMBOT. Then, the human referee performs a review and discusses eventual improvements with the data provider. Once all questions and suggestions have been satisfactorily handled, the data product is ready for step 2. The data checker will upload the latest state of step 1 to the step2 archive on the GIN. Data providers do not have access to step 2.

– For $D_{sec}$, the handling of submission differs for two reasons:
Firstly, the evaluation of optional $D_{sec}$ requires the acceptance of the obligatory $D_{min}$ for the same year. This condition is related to checking task 2.4, requiring the consistency of the two submitted data products. Thus, the data provider will be informed and receive a preliminary automatic review report. A new automatic review will be performed whenever data is modified/uploaded to step 1, and also if the $D_{min}$ reaches a new step. The human data checker however will only be informed once the $D_{min}$ is finally accepted for publication (step 3).

Secondly, the great variety of format types, versions and packing tools used to upload $D_{sec}$ to step 1, render a data review very difficult. The automatic process involves the extraction of the data structures and reformats the files into latest versions of INTERMAGNET recommended IMAGCDF archives, covering monthly data sets. This homogenization process preserves both the data contents and meta information. The data files are automatically uploaded to step 2 and newly generated whenever updates on step 1 occur. The step 1 review process of $D_{sec}$ finishes when the human referee
is uploading a review report to step 2.

## 3.2 Step 2 - the editorial task

Step 2 can be described as the editorial task of the review process. Whenever $D_{min}$ is uploaded to step 2 or when a final review report is uploaded to the one-second step 2 archive, IMBOT, continuously scanning step 2 directories, will automatically inform data providers and the chairs of the INTERMAGNET definitive data committee (IM-DD) that the main review process
is completed. The chairs of IM-DD will read review reports and eventually cross-check the evaluations. If this assessment confirms that all quality and format standards have been met, the data is approved and moved to step 3, from which it is accessible from the definitive data portals of INTERMAGNET.



### 3.3 Step 3 - the publication state

Step 3 marks the final publication stage of the review process. The transfer of data to step 3 is supported by IMBOT, which can
update the publication date within the files. Data providers are, at this point, automatically informed by mail that their data has been finally accepted and is now accessible through INTERMAGNET's definitive data portals.

## 4 IMBOT application

IMBOT runs on a Linux-based server-currently a KATOM industrial computer- hereafter denoted as the IMBOT server. The IMBOT server is maintained by an operator, the IMBOT manager, who monitors runtime and data processing workflows. The
IMBOT server accesses the INTERMAGNET GIN in Paris every day, to download any new data sets. It scans all downloaded directories for new or modified files and directories-including STEP1, STEP2 and STEP3 directories, for $D_{min}$ and $D_{sec}$ data products. New or modified files in the STEP1 directories are identified by comparing their creation and modification time with an "already processed" log. If new data is found, its directory is analyzed. Reading and writing processes are relying on the MagPy2.0 library (Leonhardt et al., 2025, 2013), which supports all data formats currently used in the geomagnetic community
and is designed to support future modifications. IMBOT consists of three separate applications:

- **IMBOT_convert:** downloads data products from the GIN and converts the directory structure of step 3 $D_{min}$, so that the step 3 directory structure is similar to step 1 and step 2, simplifying further processing.

- **IMBOT_scan:** scans step 1 folders of $D_{min}$ and $D_{sec}$, and compares file creation/modification times to a local memory, namely a json style file containing details on current states of all subdirectories.

- **IMBOT_analysis:** analyses modified data sets according to the tasks defined in section 2.

### 4.1 IMBOT for one-minute analysis

For one-minute analysis, step1 minute data is synchronized with the IMBOT server. New or modified datasets will be identified by comparing the current directory contents with a local memory from the last check. If any updates are detected, an initial read test on all files is performed using the MagPy package. If reading is successful, all datasets are subjected to check1min
(Reda, 2021) process in a wine 32bit emulation environment on the IMBOT server. Check1min, a Windows console application, performs fundamental checks on file formats, metadata consistency, reported means, and discrepancies between files. Submitting IMOs are requested to perform this data check before submitting data and include the resulting report. Running a console application however gets more and more complicated for data suppliers as such routine is not inherently supported by any modern operating system. Thus, an automatic application simplifies the future usage until INTERMAGNET updates
its format requirements for $D_{min}$. The check1MIN process verifies end-of-line characters in text files and header information, words W01-W16, in the INTERMAGNET Archive Format (IAF) binary files. It tests annual mean consistency, comparing yearmean.imo with values calculated from $D_{min}$ in IAF files. Discrepancies are only flagged if they exceed the file resolution





(1 nT & 0.1 minute). Baseline metadata and observatory metadata are checked in the imoyyyy.blv file and in the readme.imo file, respectively. While there is no INTERMAGNET specification for this file, its metadata should remain consistent with other files. Daily and hourly mean consistency in IAF files is tested, ensuring differences do not exceed 0.2 nT. The format of the yearmean.imo file is also verified. Note that check1min does not detect incorrect field formats, such as "2019 500" or "2019.500" in yearmean.imo. If check1min reports the annual values as 999999.0, it indicates insufficient data is available for a complete mean calculation (<90% of values available). For $D_{min}$, IMBOT reports results at two levels: success or failure . A failure occurs when data is not readable and check1min could not be applied.

## 4.2 IMBOT for one-second analysis

For one-second analysis, new datasets are automatically downloaded and eventually extracted (supported formats are zip, gz and tar) to a temporary directory on the IMBOT server. All data files are loaded and, subsequently the evaluation steps as outlined below will be performed. Finally, data will be exported into monthly IMAGCDF archive files, as requested by INTERMAGNET, and uploaded to the step 2 directory on the GIN. The full evaluation process is summarized within an individual IMBOT 1s report for each observatory. The report, eventually including recommendations on updates/fixes, is sent to the submitting institute. E-mail addresses are taken from a local e-mail repository or, if not available, extracted from the one-minute readme.imo submission. The report is written in Markdown language, which can be viewed using tools such as dillinger.io, GitHub or any text editor. If the dataset already satisfies all conditions for final evaluation, a data checker is assigned, and the IMBOT 1s report is sent directly to them, provided that an expected step 3 $D_{min}$ submission is available. All automatic processes are logged, and reports on newly evaluated data and eventual problems are sent to the IMBOT manager. Converted data files, reports, and, if necessary, a template for meta information updates, are uploaded to the step 2 directory for $D_{sec}$ on the GIN. The original submissions in step1 are kept unchanged. It is currently under discussion whether to delete step 2 content after final acceptance and transfer of data products to step 3.

## 4.3 Quality levels of automatic analysis

The automatic evaluation routine for IMBOT one-second data uses a tiered quality level system-level-0 to level-2- with level-2 being the highest possible grade. Data suppliers will get an automatic feedback whenever a new evaluation of their data is triggered by IMBOT, indicating a current level of the automatic checking routine.

### 4.3.1 Level-0: Critical issues

Level-0 Indicates significant problems with the data structure that may include:

- Large gaps

- Unreadable files

- Uninterpretable file structure





Institutes receiving a level-0 report must correct the issues detected and may ask the IMBOT managers for support.

### 4.3.2  Level-1: Basic Acceptance

Any uploaded complete and fully readable dataset that can be converted to an IMAGCDF format is automatically assigned to level-1. The uploaded datasets can be either IAGA-2002 files or IMAGCDF files. Supported compressed formats are .ZIP, .GZ and .TAR. If the dataset does not qualify for level-2, a file called **level1_underreview** will be created containing information on the evaluation status. It indicates that the dataset could reach the next evaluation level if appropriate information is provided or data checking is completed. A report will be sent to the submitting institute by e-mail. The data supplier is asked to solve the

listed issues in order to reach level-2. The most common issue preventing a level-2 classification is missing meta information. Data providers should use the "meta_IMO.txt" file received along with the report, add to it any missing meta information as outlined and described (an example is given in the appendix) and upload the meta_IMO.txt file to the step1 upload directory. Uploading this file-or any new data file will trigger an automatic re-evaluation.

It is possible to submit a meta_IMO.txt file with the original submission. If IAGA-2002 files are submitted, some required

information for creating INTERMAGNET CDF archives will always be missing. Supplying the metadata directly with the submission, enables reaching level-2 grades without further updates.

### 4.3.3  Level-2: Acceptance

Level-2 requires that all meta information is provided, including information on standard levels as outlined in the IMAGCDF format description, like timing accuracy, instruments noise levels etc. Besides, a level-2 check includes some basic test on

data content (completeness, time stamping etc) and includes a basic comparison with submitted/accepted $D_{min}$ to confirm its definitive character. If successful, a IMBOT 1s report is constructed (e.g. **level2_underreview.md**). Again, data providers will receive a complete report. As soon as the $D_{min}$ dataset is accepted, $D_{sec}$ is reevaluated and the data checker is assigned for the final evaluation. All acceptance tests are performed automatically by IMBOT.

### 4.4  Thresholds for acceptance levels

Regarding files, formats and meta information tests are performed as described in sections 2.1 and 2.2. Any deviation will lead to a downgrade from level-2 to level-1. If files are not readable then level-0 is assigned. If individual data points are missing (task 2.3), the report will contain the amount and month of occurrence (level-1). If more than just individual points are missing, the data set might be classified as level-0, as such missing-data-observation might be caused by corrupted uploads and downloads. The submitting institute, however, can confirm the unavailability of such data easily by using the meta_IMO.txt

template. If $F$ values are provided, IMBOT tests whether these values are independent measures of the field ($S$), as requested by INTERMAGNET. This test is done by calculating the average $\delta\bar{F}$ ($= F_{vector}$ - $F_{scalar}$) and its standard deviation ($\sigma_{\delta\bar{F}}$) on a monthly basis. If $\delta\bar{F} < 1$pT and $\sigma_{\delta\bar{F}} < 1$pT, then the provide $F$ values have very likely been calculated the vector data. If both test values exceed 10pT differences, then $F_{scalar}$ is assumed to be independent from vector data and is denoted $S$. The





report will contain a corresponding message. Temperature columns are also read and monthly mean temperatures are listed in
the report for a quick validity check.

For consistency, task 2.4, a difference analysis is performed by filtering $D_{sec}$ to one-minute ($D_{sec}^{min}$), using the IAGA/INTER-
MAGNET recommended gaussian filter (Jankowski and Sucksdorff, 1996). Then $D_{sec}^{min}$ is compared to the already accepted
$D_{min}$ on a monthly basis. Three quantities are compared from the difference of $D_{sec}^{min}$ and $D_{min}$, hereinafter referred to as
$\Delta\bar{D}$:

1. the average monthly difference for each component $\Delta\bar{D}(x,y,z)$ should be close to zero. Only in this case $D_{sec}$ and
   $D_{min}$ record the same average geomagnetic field.

2. the standard deviation of the difference $\sigma_{\Delta\bar{D}}$ also needs to be close to zero. Large $\sigma_{\Delta\bar{D}}(x,y,z)$ indicate short term
   discrepancies between $D_{sec}$ and $D_{min}$.

3. individual maximal amplitude differences, provided are extremes of differences for each component ($\Delta D_{ext}(x,y,z)$),
   help to identify differences in outlier treatment of both data sets.

If $\Delta\bar{D}(x,y,z)$ exceeds 0.3 nT then a notification for the data checker is added into the report. So far only one of all tested
records of the last 10 years exceeded this threshold. If $\Delta D_{ext}(x,y,z)$ are very small i.e. $<= 0.1$ nT which is the resolution
of $D_{min}$, this excellent agreement indicates that obviously $D_{sec}$ is the primary analyzed signal of the submitting institute,
and all "cleaning" has been performed on this data set. $D_{min}$ is then usually just a filtered product of $D_{sec}$. Slightly larger
$\Delta D_{ext}(x,y,z) <= 0.3$ nT will be observed, if either independent cleaning has been performed, an non-gaussian filter has
been used, baseline treatment differs slightly, or different instruments are the basis of both data sets. All $\Delta D_{ext}(x,y,z) <=$
0.3nT are termed excellent/good within the report and point to exemplary data treatment. If $\Delta D_{ext}(x,y,z)$ exceed 0.3nT
but are within the IM threshold of 5nT then the report will mention "small differences" within the monthly reports. Only
if peak amplitudes exceed 5nT, a notification for the human data checker will be added into the report to verify these "large
differences". The level rating remains unaffected. IMBOT is not performing tests whether data is consistent with the underlying
analysis methods like filter types, baseline methods etc, as this is not easily possible for $D_{sec}$ submissions. Consistency with
expectations by comparing with nearby sites has usually already been performed for the already accepted $D_{min}$.

A data quality assessment, as summarized in task 2.5, is performed but is not used as a criteria for level classification. IMBOT
runs tests and provides a summary of its results within the report, so that the submitting institute as well as the data checker get
feedback about quality parameters. $\delta F$ variations are calculated on a monthly basis for task 2.3, in case such data is provided
along with the data set. The average monthly $\delta\bar{F}$ is expected to be close to but not identical to zero, as shown above. Large
deviations however, exceeding $\delta\bar{F} > 3$nT indicate severe errors either in baseline adoption or data treatment. The threshold
of 3 nT is chosen, as this would indicate a single day with 90 nT deviation or half a month with 6 nT deviation, thus values
significantly exceeding baseline and other underlying data variations within the 5nT threshold of INTERMAGNET. Exceeding
the $\delta\bar{F}$ threshold will lead to a level-1 reduction. An example of such an observation is shown in section 5.

From every month, three daily records with minimal average $K_{FMI}$ (Sucksdorff et al., 1991) are extracted, altogether
36 days every year. These 36 records correspond to 10% of the collection and are then used to estimate the average noise



level. The power spectral density (PSD) of the selected daily records is calculated and the individual noise level of each selected day is obtained as the mean of the amplitude spectrum between Nyquist and a period of 10 seconds. Calculation makes use of the default matplotlib.mlab psd method (Hunter, 2007). All daily noise levels are collected and extreme outliers are removed by testing the median of distances from the median, corresponding to a 2-sigma selection in case of a normal distribution. The remaining median noise level and its uncertainty measure are provided in the report. As noise level is part of the requested StandardLevel description of the IMAGCDF's meta information, you will get some recommendation for IMOS-11 (see IMAGCDF).

## 4.5 The report

The IMBOT analysis report is designed to assist data providers and data checkers in evaluating and improving data submissions. The data checker will receive the same report as the data supplier. The IMBOT report is titled with IMO code and level assignment, it is structured into seven main sections, each aimed at addressing different aspects of data quality:

1. *Issues to be clarified for level-2.* This section contains remarks on improvements if the data product is on level-0 or level-1. It contains a list of issues and the months in which they were found.

2. *Possible improvements (not obligatory).* This section provides suggestions for the submitting institute which they might find useful.

3. *ImagCDF standard levels* as provided by the submitter, provides a list of the mandatory IMAGCDF StandardLevels, their descriptions and how the data supplier meets these criteria. In some cases, validity is verified by IMBOT and adds notes accordingly, i.e. IMOS-11 - noise level.

4. *To be considered for final evaluation.* This section is intended for datacheckers. A couple of notes are summarized, header differences which cannot be classified by the automatic process and some quality verification requests are listed.

5. *Provided Header information*, offers an overview on important header contents.

6. *Basic analysis information* lists important values as obtained from the analysis and underlying software products used for evaluation.

7. *Details on monthly evaluation* is the longest and most detailed part, it provides month-by-month summaries, including the average values and test results as discussed in section 4.2.2.

## 4.6 Homogenizing data products for publications

Every successful analysis of step 1 data (submission state) obtaining an IMBOT level-1 or level-2 is converted into a step 2 data product (editorial state). These step 2 archive files follow the naming and content conventions of the most recent IMAGCDF version1.3 (at the time of writing this article). The data checker can access the editorial step 2 files without compression and format issues. All INTERMAGNET software products recommended for data checkers can handle editorial step 2 contents.





The conversion routine also incorporates manually provided meta information from the meta_IMO.txt templates. This procedure ascertains that final publication products are standardized and easier to review, as basically all format issues have been

solved this way. Conversion, however, does not alter or modify the submitted data in any way. To illustrate this, Figure 1 shows an example of a step1 data set (top), the same day extracted from step 2 (middle) and the difference of both (bottom), demonstrating that data remains the same considering the full resolution of the original data set.

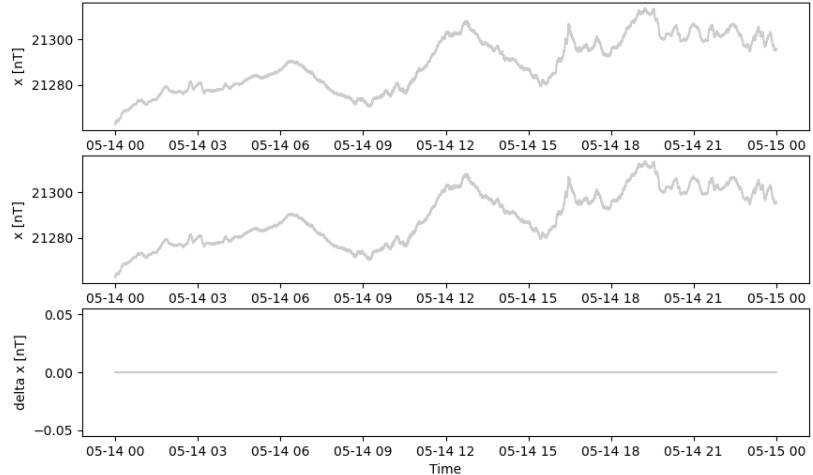

**Figure 1.** Comparison of step1 and step2 data contents, and its difference (bottom).

Step 2 files will also contain any auxiliary dataset like temperature or scalar data in the original resolution. The step 1 meta information will be preserved completely, adding any additional information provided with a metadata template. The

filename, time column names, and the numerical input types (strings will be transformed to floats) might be converted to meet the IMAGCDF 1.3 standard. Obviously, the file format type will be updated to 1.3 for step2 products.

## 5    Results for automatic analyses of step 1 data

The analysis of $D_{min}$ is relatively straightforward and the automatic routine is basically only a notification system. The check1min routine is well-established and the automatic testing and notification procedure does not contain any significant

obstacles. Therefore, this section focuses on the much more heterogeneous and voluminous $D_{sec}$, which is also the main motivation for developing an automatic assistance system like IMBOT. For the following in-depth analysis, we selected submissions from two recent years and will summarize the current status as of April 2025. It is important to note that this status is not fully reproducible as IMOs continuously review and modify their step 1 data products in order to meet publication criteria. This particularly affects 2022 data sets which are currently handled by INTERMAGNET data checkers and will gradually be

extended to earlier submissions. This manuscript is based on the submission state as of 23. March 2025.



| year | $N_{sub}$ | $N_{aut}$ | $N_{ac}$ |
|------|------|------|------|
| 2014 | 41 | - | - |
| 2015 | 44 | - | - |
| 2016 | 45 | - | - |
| 2017 | 45 | - | - |
| 2018 | 53 | - | - |
| 2019 | 50 | 40 | 4 |
| 2020 | 43 | 35 | 8 |
| 2021 | 32 | 28 | 4 |
| 2022 | 29 | 26 | 2 |
| 2023 | 20 | 14* | 0 |
| 2024 | 0 | 0 | - |

**Table 1.** Current submissions and their review state. IMBOT is running only for submission from 2019 onwards. The human cross-checking process is currently beginning. The relatively low number of level-2 data ($N_{aut}$) compared to submitted data $N_{sub}$ for 2023 is related to yet missing meta files from one organization.

## 5.1 One-second data submissions and status of automatic analysis

Table 1 summarizes the current submission status of $D_{sec}$ since the official start in 2014. Such an overview is of particular interest for data managers and although this just represents a current status at the time when submitting this article, such information can be easily extracted anytime from IMBOT using its management interaction tool **IMBOT_report**. A peak in submission ($N_{sub}$) was reached for 2018 with data sets from 53 INTERMAGNET observatories, indicating that about half of the IMOs are ready to provide such high frequency products. Shown are also the amount of successful automatic analyses in step2 ($N_{aut}$). An automatic analysis is termed successful if level-2 is reached. Only in this case, provided that corresponding $D_{min}$ has been accepted, human referees are informed and continue the evaluation process. Automatic IMBOT analyses are currently active for 2019 onwards, although earlier years have been partly analyzed for testing purposes. The amount of data sets which have been checked by human data checkers and (in all cases) have been finally accepted for publication is shown in column $N_{ac}$. Please note that the manual data checking procedure has started only recently, explaining the relative low number of currently accepted data products. In order to save storage space on GINs it is also planned to remove accepted step2 $D_{sec}$ after this data sets are published on the INTERMAGNET portal, as the underlying data will be identical. Thus, only the originally submitted raw data product and the homogenized published archive are preserved, the latter including review protocols of IMBOT and the human data checker.

## 5.2 In-depth analysis of submitted data sets

Any automatic analysis system requires accurate monitoring and statistical analysis tools particularly when it comes to preselections of data products suitable for a final review or not. This is of essential interest not only for the data publisher, but also for the data supplier who usually denotes a significant amount of work to get their data products ready for publication. In




order to outline how IMBOT is approaching these two challenges we will firstly have a detailed look on INTERMAGNET data submissions. For this report we will focus on submissions from two years, 2016 and 2022.

One-second data products from 45 observatories have been submitted for 2016. These data files have been uploaded to the step 1 folder of the Paris GIN in various different ways and formats. A summary of the underlying formats is shown in Figure 2. Submissions make use of either the IAGA-2002 format or different versions of IMAGCDF (Bracke, 2025). IAGA-2002

submissions cover daily records which then have been packed into either daily, monthly or yearly zip files using zip or tgz compressions tools. In three cases complex non-standard compression routines were used. IMAGCDF file submissions consist mostly of daily files, compressed in gnuzip or zips or just combined into tar archives. Monthly IMAGCDF files without any additional compression as requested in 2016 by INTERMAGNET are provided by 11 observatories only.

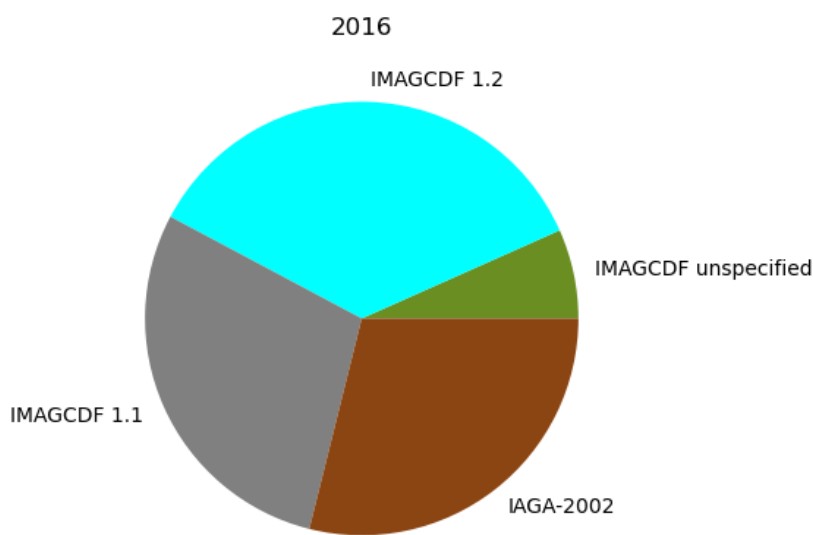

**Figure 2.** Distribution of data formats submitted in 2016. Later submissions mainly used the IMAGCDF1.2 format. About 1/3 of the submissions still use IAGA-2002 with limited 10pT resolution.

Since 2016 INTERMAGNET and several observatories provided new tools for IMAGCDF export and also the official CDF

tools improved. When looking at the submission status for 2022, the proportion of correct submissions using a modern version of the IMAGCDF format increased strongly (Figure 3). The heterogeneity in packing and compression algorithms significantly decreased. Three observatories submitted data sets with flagging information, denoted with a preliminary, in-official version number 1.3, which will hereinafter denoted as version 1.2.1. For three observatories, a step 3 $D_{min}$ is not available (ABK, DED, HRN). Therefore, these data sets are missing in the latter quality analyses.





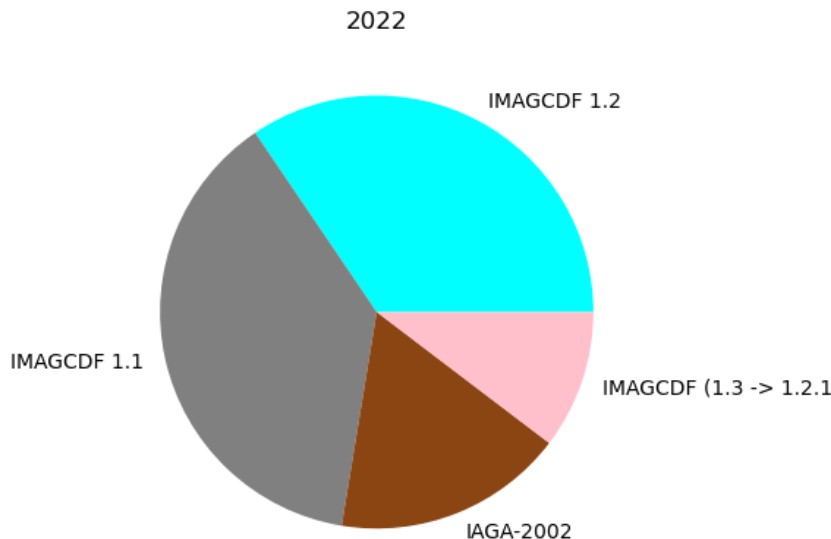

**Figure 3.** Distribution of data formats submitted in 2022. Only a few IAGA-2002 submissions remain. Some submissions use an experimental IMAGCDF version including flagging information.

The automatic analysis routine IMBOT is able to extract data from all compression and archiving formats used so far in data submission. It further can handle all different file formats and their underlying versions and data coverages. Thus the automatic system is able to overcome problems of data suppliers to fulfill stringent format requirements, which is of significant help for some institutions. Adept quickly to such requirements requires manpower and IT support which is not equally available in observatories.

## 5.3 Data quality of 2016 and 2022 submissions

After downloading and extracting all data submissions, data is subject to the checking procedure as outlined in section 4. When looking at the automatic level assignment and comparing submissions for 2016 and 2022 one can easily spot that the relative amount of level-2 grades significantly increased between those years. The main reason of level-1 grades in pre-2020 submissions is missing meta information, particularly the required information on StandardLevel classification of field PartialStandDesc ((Bracke, 2025)). For the 2022 submissions, IMBOT sent meta-information templates to the data suppliers in case such missing meta information. These meta files have been correctly used by all those observatories. The data supplier does not need to recreate all data sets, they just need to upload the required meta-information in a simple text file which will then be considered by IMBOT and included into the converted step 2 editorial data products. Thus the relative proportion of level-2 data sets is much higher for 2022 than for 2016 (Figure 4). For 2022 only three level-1 data products and no level-0




product oppose 26 level-2 data sets. For 2016 we find 21 level-1 and two level-0 data sets together with 18 level-2 data sets. The main reason for level-0, only observed in old 2016 submissions, are missing data or individual unreadable, likely corrupted data files.

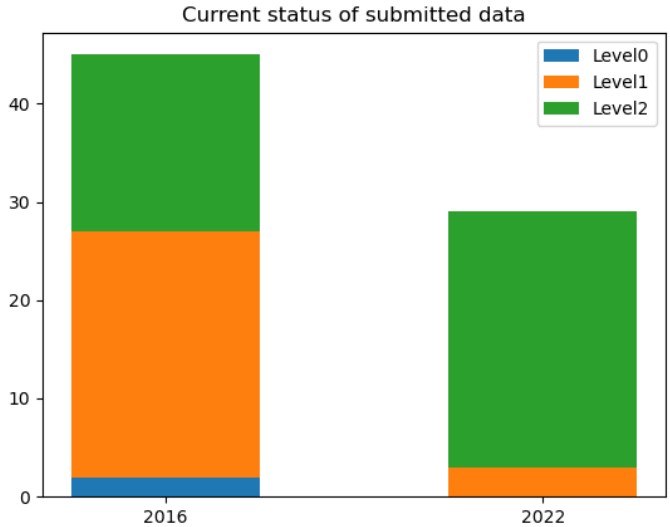

**Figure 4.** IMBOT level assignments during the automatic data checking process. In 2022 the majority of data sets fulfills all testing criteria.

After missing meta information, the second reason for level-1 products in 2016 is missing data due insufficient file coverage. The main reason for two level-1 data products in 2022 is related to significant disturbances of $\delta\bar{F}$ (see section 4.4), typically

observed for a single month only. Significant deviations of $\delta\bar{F}$ from zero, as found for two days in Figure 5, indicate a problem with baseline adoption.

For most IMOs providing $D_{sec}$, the noise level is usually relatively low, even below 20pT/$\sqrt{Hz}$ for half of the data suppliers for 2016. For 2022 the majority of submissions are characterized by noise levels below 20pT/$\sqrt{Hz}$ (Figure 6). Only two data sets exceed a noise level of 100pT/$\sqrt{Hz}$ which then needs to be mentioned in IMOS-11 of the PartialStandDesc, which one of

the observatories did. As the noise level is only estimated on 10% of the collection and the method might also differ from the techniques used by the data provider, such discrepancy is only reported but not used for level reduction. It is up to the human data checker to discuss a possible issue with the data supplier.

Noise level and a comparison to other observatories might be helpful when planning instrument and installation upgrades. Data suppliers and data checkers might test the power spectral density function for identifying technical and other noise

contributions in lower frequencies, which eventually can point to spurious signal contributions from other instruments or electronic devices.





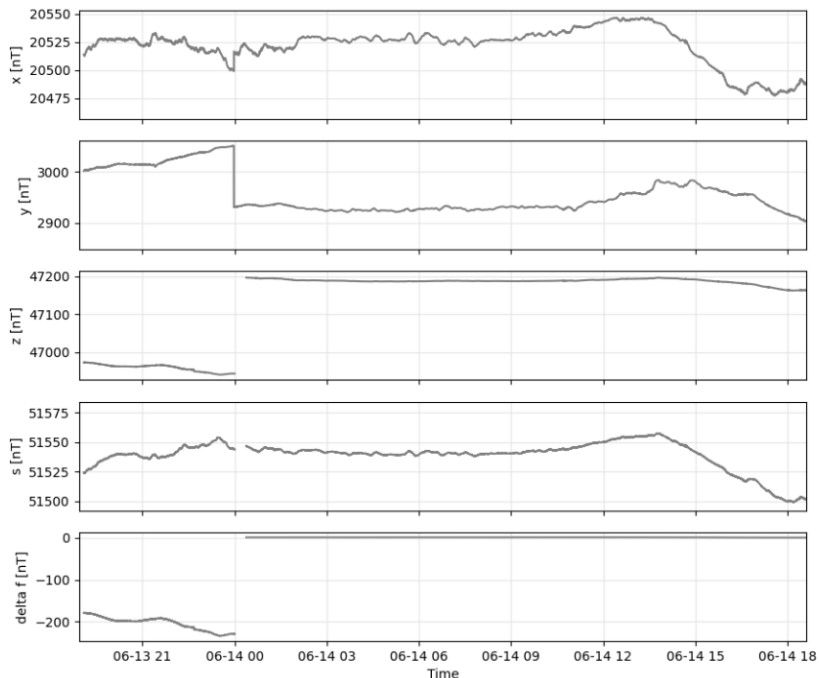

**Figure 5.** Strong deviations in delta F (G) will trigger a level reduction.

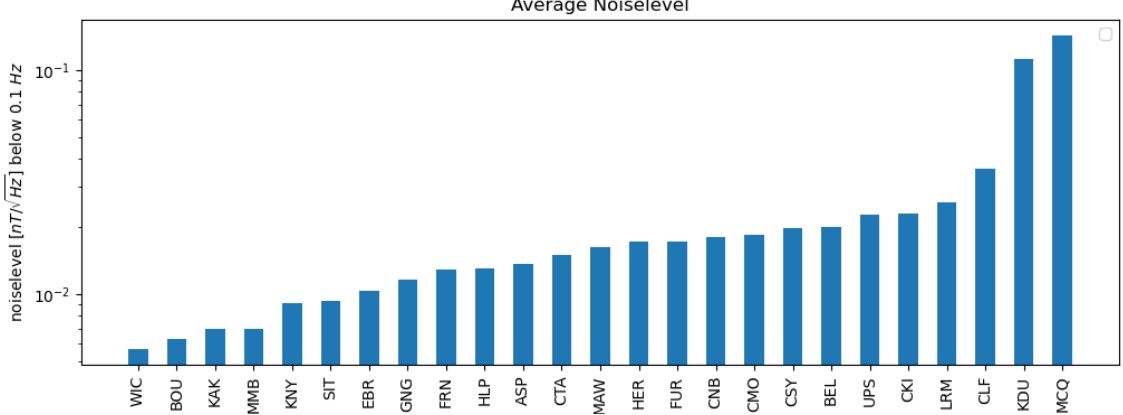

**Figure 6.** Median noise levels for all IMOs submitting data in 2022.



## 6 Discussion and Conclusion

Automatic systems for quality control are of particular interest in the evaluation of large data sets. Geomagnetic data is particularly challenging due to its non-stationary character and the highly dynamic, non-periodic signal contributions comprising
a wide range of different frequencies and origins. Careful control by the data providers and removal or marking of spurious signals is of great interest for the end-user. INTERMAGNET asks their data suppliers to perform careful data quality control, and subjects submit data to a peer review system, which unquestionably increases trustworthiness for the end-user. However, such testing and review procedures delays the data availability to end-users and increases the amount of work for referees, usually volunteers with limited time. As shown in this work, the IMBOT system is automating the evaluation process, and
informing both data providers and referees at every stage, from submission to final publication. IMBOT enhances efficiency, ensures adherence to INTERMAGNET standards, and delivers high-quality geomagnetic data to the scientific and operational user community in a timely and transparent manner. In all level-1 and level-0 cases, the automatic report contains instructions for the data supplier on how to obtain a level-2 data product. In most cases this requires meta information, re-uploading of individual files or confirmation/correction of missing data. The report will contain the respective month, so that the data sup-
plier can quickly identify the source of the error report. When updating the corresponding files and uploading the corrected data sets, IMBOT will be triggered and the data product will be reanalyzed. Therefore, the data supplier will get again an immediate feedback on his data submission and can react to eventually arising problems within hours to days, significantly speeding up the publication process . IMBOT tests various typical problems and produces an editorial output conforming with IM standards, the amount of work for data checkers is also drastically reduced.

IMBOT is ready for a number of future challenges. It can treat data which includes flagging information. IMBOT one-minute is already capable of reading and analyzing other $D_{min}$ formats i.e. like a yearly IMAGCDF one-minute data file (IMO_2016_PT1M.cdf). At the current state only basic read tests, verifying correct data formats and general readability are performed. This one-minute test module can, however, be extended for more intense data checking similar to check1min. Although IMBOT has been created for definitive $D_{sec}$ it can also be modified and used for other data sets as well. A possible
application would be high resolution variation data which could be quickly checked with such routine and provided as a tested data product by INTERMAGNET basically on the fly. Further data sources might also be included. IMBOT is written completely modular. Each checking technique is described and coded in an individual module. Thus, IMBOT can be simply extended or modified towards other tests and data sets.

It is an ongoing discussion which criteria and thresholds are necessary in order to evaluate submitted data sets. IMBOT
makes use of a conservative approach. The highest automatic grade requires that the data sets are readable, complete and (correctly) contain all requested information for the IMAGCDF file format. Data quality is tested but only acts as criteria for severe deviations from expected ranges (i.e. Figure 6).

Any final judgement of data quality or more sophisticated analysis of its definitive character is currently subject of a final analysis by a human data checker. As the data sets are automatically converted to a common data format, further data access
is straightforward. A detailed level report allows to judge the classification also for end users and eventually select data which





suit their needs. Due to the detailed standard level description of the IMAGCDF format, a level-2 product already contains essential details on data quality as provided by the data submitter. Part of this information is cross checked by IMBOT (e.g. noise level). Based on this information already a level-2 data set is complete, conclusive and usable for end users. From a modeler's perspective, this information is sufficient to work with the data products.

IMBOT can be used instantly for all future processing of new $D_{sec}$ uploads. It can also be used to start an evaluation of all submitted data sets from 2014 onwards and provides the possibility to get these data sets published on INTERMAGNET within a short time. For testing the capabilities of IMBOT and for reviewing its methods, it is possible to run IMBOT for selected observatories and to send reports and mails only to a selected group of referees. It is our intention to describe IMBOT and all methods as best as possible. The source code is accessible and, thus, the evaluation process is transparent for both submitters

and end users (https://github.com/geomagpy/IMBOT).

*Code availability.* The full code of IMBOT will be published as public repository on GitHUB. The repository is available here https://github.com/geomagpy/IMBOT, currently on private state.

## Appendix A:  Appendix - the meta_IMO template

```
## Parameter sheet for additional/missing metainformation
## ----------------------------------------------------
## Please provide key – value pairs as shown below.
## The key need to correspond to the IMAGCDF key. Please
## check out the IMAGCDF format description at INTERMAGNET
## for details. Alternatively you can use MagPy header keys.
## Values must not contain special characters or colons.
## Enter "None" to indicate that a value is not available
## Comments need to start in new lines and every comment line.
## must start with a hash.
## Please note – you can also provide optional keys here.
##
## Example:
# Provide a valid standard level (full, partial), None is not accepted
StandardLevel  :   partial

# If Standard Level is partial, provide a list of standards met
PartialStandDesc  :   IMOS11,IMOS14,IMOS41
```



```
     # Reference to your institution (e.g. webaddress)
     ReferenceLinks  :  www.my.observatory.org
     # Provide Data Terms (e.g. creative common lisence)
     TermsOfUse  :  Do whatever you want with my data

     # Missing data treatment (if data is not available please uncomment)
#MissingData  :  ignore
```

*Author contributions.* RL had the initial idea, has written the python package and most of the manuscript, and performed all analyses therein. BM, TM and JR contributed to the python package by discussing and defining its testing environemnts and reports. BM and TR have contributed to the manuscript by writing one-minute checking tasks. JR has written the one-minute testing tools which are part of the IMBOT package and contributed to the manuscript.

*Competing interests.* There are no competing interests.

*Acknowledgements.* We would like to thank all INTERMAGNET data checkers for their extremely valuable work and many discussions which actually resulted in this manuscript. We particularly profited from many discussions with Sergey Khomutov.



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
