# Peer review of "Peer-review of data products: an automated assistance system for INTERMAGNET"

_EGUsphere, 2025_

## Author Response (AR1)

**Response to the comments of the referees:**

**Ref 1:**

*Dear referee 1: thank you very much for your review and the constructive comments you will find our reply inline just below your comments in italic font. DONE indicates that we included your corrections into the revised version of the manuscript as suggested.*

This is a relevant manuscript that addresses the task of ensuring the delivery of high-quality data from geomagnetic observatories to the end user, and how a number of those tasks have been automated by IMBOT, the INTERMAGNET ROBOT, which has been developed to perform automated routines and quality controls that would otherwise have to be carried out unselfishly by (human) data checkers, thus speeding-up the process. I believe that IMBOT is an outstanding tool and I assume that the work behind it is enormous and deserves recognition, and this manuscript is a good opportunity to do so.

For this reason, and because the presentation is well structured and clear, I recommend the publication of this manuscript, essentially in its current form, except for a few technical issues or typos, as described below. My only "minor comment", also described below, is the addition of a diagram or flowchart to summarize IMBOT's tasks, which could replace current Figure 1.

**Technical/minor comments:**

L153: "server-currently": replace the hyphen with a long dash, otherwise it can be confused with a compound word.

*DONE*

L170: I guess the acronym "wine" should be capitalized (at least the first letter). Please check.

*DONE*

L175 vs. L170: check1MIN vs. Check1min. Please, cHecK for consistency between capital letters.

*DONE, checked all other occurences as well*

L178: Use a different format for the wildcard characters yyyy in "imoyyyy.blv", e.g., italics or bold.

*DONE*

L218: file-or: remove the hyphen.

*DONE*

L237: the provide**d** F values have very likely been calculated **from** the vector data.

*DONE*

L269: Add a space between 5 and nT.

*DONE, added a space for all occurences*

Fig. 1: The fact that the data in step 2 has not changed with respect to the data in step 1 does not need to be supported by a figure. It is perfectly clear from the text. However, I would strongly recommend adding a diagram or flowchart representing IMBOT's tasks. I believe this will help the

reader to gain an overview of IMBOT's usefulness in relation to the manual process, while also supporting and summarizing the description of the tasks carried out in section 4.

*Agreed. We added a flowchart depicting the basic data checking procedure. We included that figure in section 4.2 as it shows the procedure for one-second data checking which is slightly different to one-minute. Basically all examples in the manuscript are related to one-second data checking. We also removed the previously contained figure 1 with the zero differences, as suggested by the referee.*

L320: Remove the dot after 23.

*DONE*

L352: in-official or unofficial?

*DONE*

L358: Adept or adapt?

*DONE*

L366: … in case such **meta-information was missing**.

*DONE*

L373: … due **to** insufficient …

*DONE*

Fig. 6: I'm probably missing something, but there are 25 observatories in the x axis, while a total of 29 observatories provided data in 2022 according to Table 1. Is this because ABK, DED and HRN have been removed (as specified in L353)? But anyway, there is still one missing observatory(?).

*DONE. Thanks for recognizing this issue, this was indeed a mistake. The correct plot contains the noiselevel of 28 observatories. We do not distinguish between level1 and level2 here. The only missing observatory is Deadhorse (DED) of which one-minute data has not been accepted and thus the term „definitive" one-second data is not applicable. We added the new plot and a note in the figure caption.*

Final remark for the Conclusions: Would the authors dare to give a figure for the amount of work that IMBOT saves human data checkers? This would help the reader to understand the importance of this tool. For example: "since its implementation, IMBOT has reduced the workload of data checkers by approximately x %". Or: "It is estimated that the work carried out by IMBOT is equivalent to x hours of human labour".

*DONE*

**Ref 2:**

*Dear referee 2: thank you very much for your review and the constructive comments you will find our reply inline with your comments. DONE indicates that we included your corrections into the revised version of the manuscript as suggested.*

This manuscript presents a timely update in the peer review of geomagnetic data products: The authors introduce IMBOT, an automated assistance system designed to streamline the validation of

both one-minue and one second data submissions to INTERMAGNET. The tool addressses a real operational need to reduce the burden on volunteer data referees while ensuring adherence to rigorous data quality and formatting standards. The manuscript is well structured, technically sound, and written clearly. The Figures in this version are all present and supports the content of the manuscript.

Major comments
* * *
1. "final judgement of data quality or more sophisticated analysis... is subject of a final analysis by a human data checker" (Section 6): The manuscript could benefit from a deeper discussion on the boundaries of automation e.g., how are final decisions handled? How is the human referee alerted, and what tools are they provided with to validate these?

2. Given that the IMAGCDF format is still evolving, how adaptable is IMBOT to new versions? Will support be actively maintained, and is there a defined update pathway for future INTERMAGNET requirements?

3. The authors mention modularity as a strength ("IMBOT is written completely modular. Each checking technique is described and coded in an individual module. Thus, IMBOT can be simply extended or modified towards other tests and data sets"). Could this system (or a forked version) be adapted to other global geomagnetic networks such as MAGDAS?

*Reply to points 1-3: All question refer to the discussion part of the manuscript. We added specific sections to the discussion which provide information on all of these questions (see lines 410ff of the revised version). We also provide now the direct link to the GitHUB repository, so that readers can directly access the repository, development branches, issues. There it is also possible to fork the repository, if interested in adapting modules or methods for other projects.*

Minor comments
* * *
1. Terms such as 'step 1', 'step 2' etc. are used but a flow diagram or table summarising these steps would be aid clarity, especially for users outside INTERMAGNET.

*DONE, see also our comments to referee 1. We added an overview graph depicting structure and flow of the data checking procedure.*

2. The appendix gives helpful guidance on metadata structure. However, a sample filled-in meta_IMO.txt template would be useful for user guidance (maybe as a figure or in the appendix).

*DONE. A couple of sample filled-in templates are provided in the GitHUB directory of IMBOT. A link to this examples is given in the appendix.*

3. Page 18 Author contributions 'testing environemnts' -> 'testing environments'.

*DONE*

**Ref 3:**

*Dear referee 3: thank you very much for your review and the constructive comments you will find our reply inline with your comments. DONE indicates that we included your corrections into the revised version of the manuscript as suggested.*

**Peer-review of data products: an automated assistance system for INTERMAGNET, Leonhardt, R. and Heumez, B. and Raita, T. and Reda, J.**

**Summary:**

The authors present a new automated method/tool, "IMBOT," designed to improve the efficiency of the otherwise high-load and time-consuming peer review of data products, with particular emphasis on the INTERMAGNET 1-sec and 1-min definitive data. The manuscript provides valuable insights for data users, quality control specialists, and geomagnetic observatory operators responsible for submitting this type of data. Overall, it is a well-written contribution with a clear workflow and a coherent structure.

**Only few minor comments:**

In general, please revise the consistency in the use of the terms step 1, step 2, and step 3, as they appear in three different forms throughout the manuscript.

I recommend including a figure or workflow diagram to illustrate the IMBOT work process and its associated tasks.

*DONE, see reply to ref 1*

**L24:** The term 1-min is used (as introduced in the abstract), but from this point onward the manuscript consistently uses one-minute. Please ensure consistency in terminology.

*DONE*

**L106**, **L258**, **L259** and **L269**: Add a space in 5nT → 5 nT.

*DONE, see also ref1*

**L257**: Add a space in 0.3nT → 0.3 nT.

*DONE, see also ref1*

**L267**: Add a space in 3nT → 3 nT.

*DONE, see also ref1*